# MicroRNA Cross-Involvement in Autism Spectrum Disorders and Atopic Dermatitis: A Literature Review

**DOI:** 10.3390/jcm8010088

**Published:** 2019-01-14

**Authors:** Alessandro Tonacci, Gianluca Bagnato, Gianluca Pandolfo, Lucia Billeci, Francesco Sansone, Raffaele Conte, Sebastiano Gangemi

**Affiliations:** 1Clinical Physiology Institute-National Research Council of Italy (IFC-CNR), Via Moruzzi 1, 56124 Pisa, Italy; lbilleci@ifc.cnr.it (L.B.); francesco.sansone@ifc.cnr.it (F.S.); raffaele.conte@ifc.cnr.it (R.C.); 2School and Division of Allergy and Clinical Immunology, Department of Clinical and Experimental Medicine, University Hospital “G. Martino”, Via Consolare Valeria SNC, 98125 Messina, Italy; gianbagnato@gmail.com (G.B.); gangemis@unime.it (S.G.); 3Department of Biomedical and Dental Sciences and Morphofunctional Imaging, University of Messina, Via Consolare Valeria 1, 98125 Messina, Italy; gpandolfo@unime.it

**Keywords:** Allergy, Autistic Disorder, Dermatitis, Genetics, Immunity, MicroRNAs

## Abstract

Autism Spectrum Disorder (ASD) is a category of neurodevelopmental disturbances seriously affecting social skills, to which the scientific community has paid great attention in last decades. To date, their pathogenesis is still unknown, but several studies highlighted the relevance of gene-environment interactions in the onset of ASD. In addition, an immune involvement was seen in a wide number of ASD subjects, leading several researchers to hypothesize a possible common pathogenesis between ASD and immune disturbances, including Atopic Dermatitis (AD). In general, among potential contributing factors, microRNAs (miRNAs), small molecules capable of controlling gene expression and targeting mRNA transcripts, might represent one of the major circulating link, possibly unraveling the connections between neurodevelopmental and immune conditions. Under such premises, we conducted a systematic literature review, under the PRISMA guidelines, trying to define the panel of common miRNAs involved in both ASD and AD. The review retrieved articles published between January 1, 2005, and December 13, 2018, in PubMed, ScienceDirect, PsycARTICLES, and Google Scholar. We found a handful of works dealing with miRNAs in ASD and AD, with the most overlapping dysregulated miRNAs being miR-146 and miR-155. Two possible compounds are abnormally regulated in both ASD and AD subjects, possibly cross-contributing to the interactions between the two disorders, setting the basis to investigate more precisely the possible link between ASD and AD from another, not just clinical, perspective.

## 1. Introduction

Autism spectrum disorder (ASD) is a heterogeneous group of neurodevelopmental disorders characterized by impairments in social interaction and communication and restricted or stereotyped interests and behaviors [1], typically occurring before the fourth year of life [2]. In the last decades, there has been an increased attention of the scientific community towards this group of conditions, with a consequent improvement in the knowledge about this topic, and contextually, a broadening of the diagnostic criteria [3,4].

Nonetheless, ASD is the most heritable among neuropsychiatric disorders, with genetic contributions accounting for around 80% of ASD risk [5,6,7,8,9], and higher risk is seen in siblings of autistic children [10]. Although epidemiological studies provide information on the genetic contribution to ASD, less is known about the putative genes involved or the frequency of specific polymorphisms and variants (single-nucleotide or copy number variants).

Whole-genome and candidate-gene analyses have shown the complex genetic background of ASD, characterized by high individual differences and variability, with many ASD-risk genes involved in synaptic plasticity and gene products modifying synaptic number and strength. In addition to inherited variants, individuals with ASD often carry de novo genetic variants, defined as variants not present in the parental genome and found for the first time in the proband (see [11] for a related original article or [12] for a review on this argument). Such mutated variants affect biological pathways involved in synaptic plasticity and connectivity at different levels. It has been proposed that the interplay of (mainly) inherited rare and (to a lesser extent) de novo variants could be pivotal in the development of ASD [13,14,15], with some genes (e.g., *NLRP2*, *MOGS*) surprisingly found to be concerned with ASD and having an established role in the immune system [13]. Indeed, in general, de novo variants have a reduced impact in ASD, explaining below 5% of the genetic variance in the liability to ASD [16].

The multigenic condition of ASD seems also to be dependent on gene-environment interactions; epigenetic mechanisms involving DNA methylation, transcriptional regulations, and post-translational changes in histone proteins, are all relevant to neurodevelopmental processes that can be affected in-utero by maternal lifestyle factors [17]. Furthermore, chemical or heavy metals exposure appears to strongly contribute to ASD development [18,19,20].

Consistently, recent studies reported the clinical association between ASD and atopic disorders, such as asthma or atopic dermatitis (AD) [21,22,23,24], strengthening the link between neurodevelopmental disorders and immune diseases. Together with this possibly speculative evidence, several works hypothesized a fundamental role for the immune system and neuroinflammation in ASD development (see [25] for a review).

In fact, immunity is thought to play a key role in the neurodevelopment of both central and peripheral nervous systems, as it regulates neuronal proliferation, synaptic formation, and plasticity, removes apoptotic neurons, and actively participates in a number of neurological processes [26,27,28,29].

Furthermore, many studies reported an alteration of immune responses in children and adults with ASD, which are more frequently subject to infections, allergies, asthma, dermatitis, and over-reactions to autoimmune disorders [27,30,31,32,33]. As such, some proteins, named alarmins, featuring multiple functions, including the activation of innate immunity and the recruitment of antigen-presenting cells stimulating an adaptive response, have been hypothesized to be suitable biomarkers of inflammation in ASD and possibly used to interfere with the immune system ameliorating symptoms of ASD [34,35].

Thus, in recent times, a clinical association between ASD and inflammatory diseases, including AD, was hypothesized [22,23], and in this regard, the investigation of a possible common genetic basis is critical for the current scientific knowledge.

As evidenced by Billeci and colleagues, AD, defined as a chronic inflammatory disease, puts patients at higher risk of developing one of more of the other atopic conditions, therefore it is considered as the beginning of the so-called “atopic march” [22].

This condition, also determined by a close gene-environment interaction [36], appears to be correlated with a number of mental health conditions, including, according to recent literature, ASD [22].

Among the compounds which could possibly explain this link, up to now hypothesized from a clinical point of view, microRNAs (miRNAs) were recently seen to play a role in several molecular and cellular mechanisms, including neurodevelopment, brain plasticity, and immunity [37,38].

The miRNAs might participate in the pathological process both in neurological conditions, including autism, and in atopic disorders, including AD [39]. The overlapping microRNAs in ASD and AD could therefore allow exploration of the role of genetics in the hypothetical common pathophysiological pathway of these two conditions.

### 1.1. General Insight into MicroRNAs

It is known that miRNAs are very short (18–25 nucleotides), single-stranded non-coding RNAs, able to control gene expression and to target mRNA transcripts, possibly bringing on their translational degradation or their repression, with particular degrees of complementarity [40]. The targeting of mRNA transcripts by miRNA occurs since one miRNA is able to target a number of mRNA transcripts; conversely, a single mRNA transcript can be targeted by many miRNAs.

Actually, miRNAs have rapidly induced a great interest in humans, being potential biomarkers for diagnostic and prognostic aims [41,42], with the number of classified miRNA increasing to over 2500 potential molecules in the *Homo Sapiens* genetic makeup [40]. Despite being small molecules not capable of encoding proteins, miRNAs hold important structural, regulatory, and catalytic functions.

The miRNA genes are located in the introns of protein-coding genes or in independent non-coding DNA loci [43], whereas nearly half of the total miRNAs are pooled on chromosomes with a common promoter [44].

The biogenesis of miRNA is extremely complex, consisting of several phases. These include: (i) in the nucleus, the transcription of miRNA genes into primary miRNA transcripts by RNA polymerase II; (ii) the freeing of pre-miRNA hairpin, through the trim of the primary miRNA transcripts by the RNAse III Drosha endonuclease; (iii) the active exportation of the pre-miRNA hairpin out of the nucleus in a process involving the nucleocytoplasmic shuttler Exportin-5; (iv) the final maturation, in the cytoplasm, processed by Dicer RNase III endonuclease, splitting the pre-miRNA into a single-stranded mature miRNA [45]; (v) the binding of the mature miRNA to proteins of the Ago family, and (vi) the assembly of the RNA-induced silencing complex (RISC) together in order to employ its physiological functions.

The mature miRNA, once incorporated into the RISC, induces post-transcriptional gene silencing by binding RISC to be partially complementary to the target mRNA found mainly within the 3’-untranslated region (UTR) [46,47].

Flaws in miRNA expression deeply affect several pathways related to cell regulation, including apoptosis, stress responses, or cell proliferation throughout the human body [48,49,50]. Indeed, a single miRNA could repress around 100 mRNAs, while around 60% of human protein coding-genes are represented by conserved targets of miRNAs, thus a number of mRNA targets are regulated by miRNAs [51].

### 1.2. MiRNAs Linked to Brain Function

MiRNAs approximately regulate two-thirds of human mRNAs [51], and are as much as 70% expressed in the central nervous system (CNS), including the brain and spinal cord [52,53]. Their changes during childhood are different depending on the affected brain region [54].

In particular, miRNAs are abundant in neurons and glia, often placed at the synaptic level, and able to regulate the structure of the dendritic spine, as happens with miR-134 which reduces spine growth by targeting spine growth-promoting kinase Limk1 [55].

Indeed, dendritic spines are bulges on a dendritic tree of a neuron, composing the post-synaptic termination of a synapse, reflecting—through their structure—the degree of brain maturation, and, somehow, brain plasticity.

Their density is abnormal in several conditions, including schizophrenia (dendritic spines loss) and ASD, the latter featuring an increase in the quantity of spines in specific brain areas [56,57,58,59].

Several other miRNAs are associated with dendritic spine structure, including miR-125b [60], miR-132 [61], miR-137 [62], and miR-138 [63].

Overall, several miRNAs affect brain functions and development, neuronal plasticity, maturation, and differentiation [37,64]. Dysregulation of miRNA expression is particularly frequent within several neurological disorders, including ASD, therefore the association between some common miRNA families and ASD, despite still largely unknown, is nowadays clearer than in the past.

### 1.3. MiRNAs and Skin Disorders

Recently, much evidence has been published about the role of miRNAs in several cellular processes, including immune response, DNA repair, apoptosis, proliferation, and differentiation [65], but also in morphogenesis, differentiation, wound healing, psoriasis, and AD [66,67,68]. Specifically, AD pathogenesis is also associated with a complex gene-environment interaction, as well as with an alteration of the skin barrier function, and a deregulation of the immune system [69]. Several miRNAs, including miR-146a, miR-155, miR-203, and miR-483–5p, are also differentially expressed in AD and in other immunologic and inflammatory disorders.

## 2. Materials and Methods

A literature review of the articles published between 1 January 2005, and 13 December 2018, was conducted in PubMed, ScienceDirect, PsycARTICLES, and Google Scholar following the PRISMA guidelines.

### 2.1. Studies about ASD

The search strategy for this part was as follows: ((“micrornas” [MeSH Terms] OR “micrornas” [All Fields] OR “microrna” [All Fields]) AND ((“autistic disorder” [MeSH Terms] OR (“autistic” [All Fields] AND “disorder” [All Fields]) OR “autistic disorder” [All Fields] OR “autism” [All Fields]) OR (“autism spectrum disorder” [MeSH Terms] OR (“autism” [All Fields] AND “spectrum” [All Fields] AND “disorder” [All Fields]) OR “autism spectrum disorder” [All Fields]))).

### 2.2. Studies about AD

In this part, the search strategy was as follows: ((“microRNAs” [MeSH Terms]) AND (“skin” [MeSH Terms] OR “dermatitis” [MeSH Terms] OR “urticaria” [MeSH Terms] OR “eczema” [MeSH Terms] OR “hypersensitivity” [MeSH Terms])).

Overall, the search was limited to articles describing studies conducted on humans published in peer-reviewed journals. After having discarded duplicates, the obtained results were sorted by relevance and the most significant works related to ASD and miRNAs and to AD and miRNAs were selected. We will first present the results from the literature review and then discuss the possible associations between miRNAs in ASD and AD according to such findings.

## 3. Results

The literature search displayed 26 articles directly related to the relevant topics (Figure 1).

### 3.1 Studies about ASD

According to the literature review, a number of miRNAs were found to be associated to ASD (Table 1). Specifically, the most overlapping dysregulated miRNAs appeared to be let-7, miR-19b, miR-23, miR-106, and miR-146.

### 3.2 Studies about AD

A few works studied microRNAs involvement in AD (Table 2). Here, the main dysregulated miRNAs are miR-146, miR-155, and miR-203.

Summarizing, the association between ASD and AD revealed a common unbalance for miR-146 and miR-155.

## 4. Discussion

### 4.1. The Overlap between Atopy and Autism

Recent data strongly support the clinical association between atopy and ASD [21,22]. It has been widely demonstrated that allergic diseases, especially food allergies, are more frequent among ASD children [95,96].

Notably, a large observational study, comparing 14,812 atopic subjects with 6944 non-atopic subjects, with no lifetime atopic disease, highlighted a strong association between atopy and the risk of developing ASD [97]. Furthermore, autoimmune disorders, including psoriasis (2-fold risk), are also frequently identified in ASD [98].

Beside the robust clinical evidence for the association between atopy and ASD, an intriguing neuroinflammatory hypothesis has been advanced for ASD, involving the disruption of the brain blood barrier induced by inflammatory molecules, brain mast cell activation, and mast cells-microglia interactions [99].

In addition, specific environmental factors, including infectious pathogens, food allergens, toxins, and toxic metals (e.g., aluminum, lead, mercury) may negatively act on neurodevelopment through the alteration of the immune response [100,101,102].

However, the hypothesis that the pro-inflammatory cascade induced by AD could lead to ASD in the presence of genetic susceptibility, is supported by the clinical association and by a shared pattern of cellular damage, with epigenetic changes as a common pathogenic mechanism.

### 4.2. Role for Overlapping MiRNAs in ASD and AD

The main literature finding concerning miRNAs in AD and ASD is represented by miR-146a. Upregulated in various neurodevelopmental disorders [82], miR-146a was reported to be highly expressed throughout the cortex, hippocampus, and amygdala, key structures for higher cognitive functioning [103]. Furthermore, it was demonstrated that reproducing abnormal miR-146a expression in mouse primary cell cultures leads to impaired neuronal dendritic arborization—producing shriveled dendritic trees with branching points at more proximal levels compared to controls, proving the defective neural connectivity typical of ASD—and to increased astrocyte glutamate uptake capacities [82], in turn modifying fast synaptic transmission at the CNS level. Further, miR-146a, expressed in the developing brain, is enclosed within neurons, with poor expression in the glial lineage in adult mice. However, it generally inhibits the expression of neuron-specific targets, including Nlgn1 and Syt1, preventing glial cells from mistakenly adopting neuron-specific phenotypes.

In addition, the neuron excitation at the cortex is probably affected by miR-146a deregulation through the involvement of potassium two pore domain channel subfamily K member 2 (KCNK2), having a key role in neural excitability and migration at the cortex level of developing mice, a critical issue in ASD.

Furthermore, miR-146a expression contributes to neuroinflammation in the brain of ASD subjects, having a role in immune system regulation.

Its function in the regulation of inflammatory processes could partially fill the gap between ASD (and neurodevelopmental disorders in general) and AD (and atopic conditions in extenso).

It is evident indeed that an increased miR-146a expression is present in the lesional skin of AD patients [104], as it inhibits nuclear factor κ B (NF-κB)-mediated proinflammatory cytokines and chemokines, bringing alleviation of the inflammation directly linked to AD and similar conditions [91].

In AD, during skin inflammation, miR-146a is increased in keratinocytes, controlling chronic inflammatory processes triggered by IFN-γ and the activation of NF-κB. Indeed, the relevance of miR-146a in inflammatory skin disorders is confirmed by evidence from psoriasis research [105]. Furthermore, the expression of miR-146a is strongly dependent on NF-κB, and the miRNA has been shown to suppress the NF-κB signaling pathway through a direct targeting of a number of compounds, including IL-1 receptor–associated kinase 1 (IRAK1), TNF receptor–associated factor 6 [106], v-rel avian reticuloendotheliosis viral oncogene homolog B (RELB) [107], and CARD10 [108].

Moreover, it was discovered that mice with a deficiency of miR-146a develop a late autoimmunity caused by an impaired activation of NF-κB in T cells, and signal transducer and STAT1 activator in regulatory T cells [109].

Of note, previous work demonstrated that both an enhanced opioidergic activity and reduced vitamin D levels could represent shared features of AD [110] and ASD [111], and possibly miR-146a and miR-155 could interact with the genetic milieu in subjects with these disorders. Indeed, both miR-146a and miR-155 have been tested in models of LPS tolerance and miR-146 was able to amplify the severity of morphine-mediated hyper-inflammation [112].

Finally, the regulatory role of miR-146a also occurs at lung alveolar epithelial cells, where the release of IL-8 and CCL5 occurs independently from IL-1β signaling.

Concerning miR-155, its role in ASD is not yet known, whereas in AD it appears to modulate T helper type 17 (Th17) cell differentiation and function [92], and to directly target the suppressor of cytokine signalling-1 (*SOCS1*) gene, taking part in a negative feedback loop to attenuate cytokine signaling [113]. Interestingly, miR-155 is also linked to inflammation and immunity, thanks to its potent upregulation in immune cell lineages, including lymphocytes, fibroblasts, macrophages, mast cells, and dendritic cells, in turn implicated in the pathogenesis of chronic skin inflammation [114,115,116,117,118,119]. Briefly, miR-155 seems to be also involved in the regulation of T-cell responses through a suppression of cytotoxic T-lymphocyte antigen 4 and by enhancing T-cell proliferation [66]. Its deregulation, seen as increased expression in peripheral CD4 T cells of AD patients, was correlated to disease severity, supporting its role in AD pathogenesis [92].

## 5. Conclusions

Beyond the clinical evidence [22,23], a possible, yet speculative, role for genetics (miRNAs in particular) can be hypothesized to justify the clinical association between AD and ASD.

However, both miR-146a and 155 appear to be involved in this common pathogenetic pathway, despite the role of the latter still being poorly known. Several other aspects differentiating these diseases remain elusive, including the identification of putative environmental injuries and the complex role of vitamin D in immune and neurologic disorders. Therefore, further studies focusing on the association between vitamin D and opiod receptors in skin and neurologic disorders should investigate the role of target genes for common dysregulated miRNAs, in order to discover specific overlapping features of these conditions.

It remains evident that an inflammatory component is active in both diseases, and the actual data support future applications for miR-146a, both as a biomarker and as a target for therapy.

Interestingly, it can be speculated that a deregulation of miR-146a occurs earlier, during embryonic development, thus participating in the development of ASD. Apart from the well-described effect on NF-κB activity and the associated inflammatory pathways strictly linking miR-146a with AD, the deregulation of miR-146a and miR-155 could influence a wide range of their validated targets (Figure 2), essential for brain development and function. However, it remains complex to determine the major source (skin vs brain) of miR-146a and miR-155, and whether they are potentiating each other or having more organ- or disease-specific effects. These aspects warrant future larger longitudinal studies. Finally, a better understanding of the link between ASD and AD might be useful to investigate whether a specific miRNA could act as a biomarker for the risk of developing ASD for patients with AD, and potentially represent a target for ASD prevention.

## Figures and Tables

**Figure 1 jcm-08-00088-f001:**
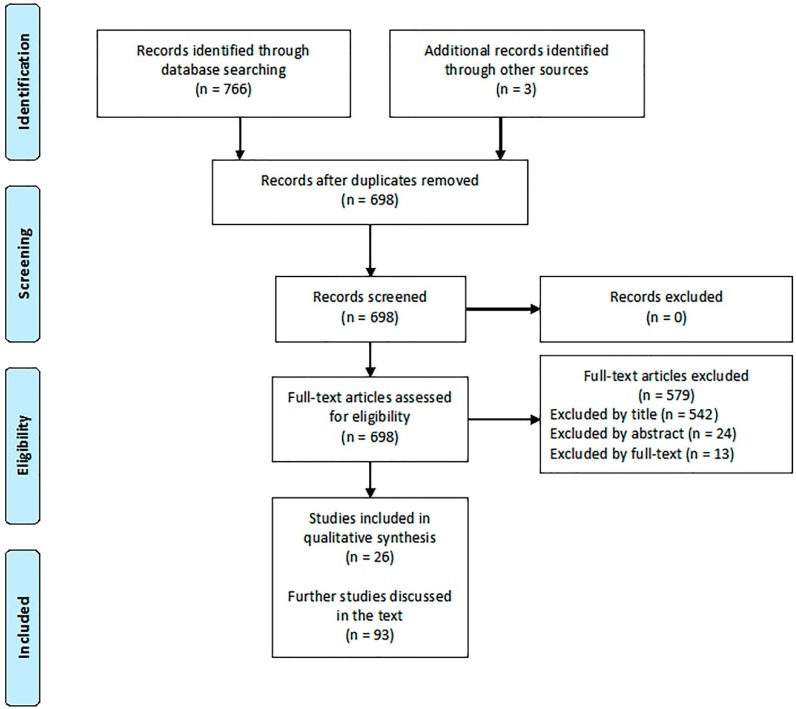
Study selection.

**Figure 2 jcm-08-00088-f002:**
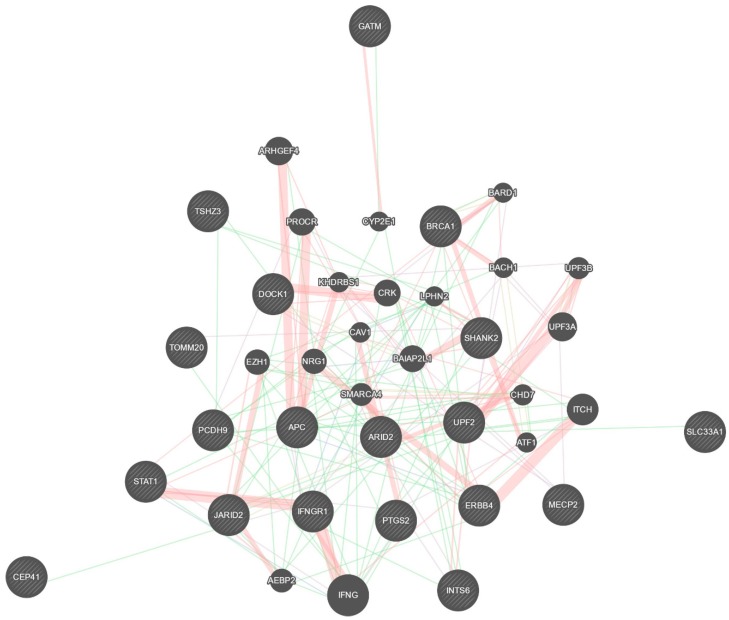
Validated targets for miR-146a and miR-155.

**Table 1 jcm-08-00088-t001:** MicroRNAs directly involved in autism (*: see explanation in the following column).

Study	*N* (Case/Control)	Design	Findings	
			Up-Regulated miRNA	Down-Regulated miRNA
Abu-Elneel et al. (2008) [70]	26 (13/13)	Measure of the expression level of 466 human miRNAs from postmortem cerebellar tissue by multiplex real-time PCR, with 377 miRNAs detected and used for further analysis	miR-106a, miR-106b, miR-140, miR-146b, miR-181d, miR-193b, miR-320a, miR-381, miR-432, miR-539, miR-550, miR-652	miR-7, miR-15a, miR-15b, miR-21, miR-23a, miR-27a, miR-93, miR-95, miR-128, miR-129, miR-132, miR-148b, miR-212, miR-431, miR-484, miR-598
Sarachana et al. (2010) [71]	14 (5/9)	Lymphoblasts derived from peripheral lymphocytes were obtained; miRNA expression profiling performed by high-throughput miRNA microarray analysis. Differentially expressed miRNAs confirmed by qRT-PCR analysis, putative target genes of two of the confirmed miRNA validated by knockdown and overexpression of the respective miRNAs	miR-16-2, miR-106b, miR-132, miR-133b, miR-136, miR-139, miR-148b, miR-153, miR-182, miR-189, miR-190, miR-199b, miR-211, miR-219, miR-326, miR-367, miR-455, miR-495, miR-518a, miR-520b	miR-23a, miR-23b, miR-25, miR-29b, miR-30e, miR-93, miR-103, miR-107, miR-185, miR-186, miR-191, miR-194, miR-195, miR-205, miR-342, miR-346, miR-376a-AS, miR-451, miR-519c, miR-524
Talebizadeh et al. (2008) [72]	12 (6/6)	Six subject with Autism Spectrum Disorders (ASD) (3 males, aged 5, 12, and 14 years, and 3 females, aged 6, 11, and 13 years), 6 age- and gender-matched TD controls. Lymphoblastoid cell lines, quantitative PCR,	miR-23a, miR-23b, miR-132, miR-146a, miR-146b, miR-663	miR-92, miR-320, miR-363
Mundalil Vasu et al. (2014) [73]	110 (55/55)	55 ASD (48 males, 6 females, aged 11.29 ± 5.45 years), 55 TD controls (41 males, 14 females, aged 11.3 ± 2.37 years). RNA extracted from serum, mature miRNAs selectively converted into cDNA. The expression of 125 mature miRNAs was compared between pooled control and ASD samples. The differential expression of 14 miRNAs further validated by SYBR Green quantitative PCR of individual samples. Target genes and pathways of miRNAs predicted by DIANA mirPath software	miR-19b-3p, miR-27a-3p, miR-101-3p, miR-106-5p, miR-130a-3p, miR-195b-5p	miR-151a-3p, miR-181b-5p, miR-320a, miR-328, miR-433, miR-489, miR-572, miR-663a
Popov et al. (2012) [74]	55 (30/25)	Thirty ASD (24 males, 6 females, aged 3–20), 25 TD controls (20 males, 5 females, aged 3–20 years). Whole blood collection, analysis of gene expression changes applying LC expression profiling service, using pooled whole blood-derived total RNA samples		miR-486-3p
Seno et al. (2011) [75]	42 (20/22)	20 severe ASD (13 males and 7 females), 22 unaffected siblings (19 males and 3 females). Lymphoblastoid cell lines, RNA was extracted and assayed using Illumina gene and miRNA expression arrays. Control quality in BeadStudio (Illumina)	miR-10a, miR-30a, miR-181a, miR-181b, miR-181c, miR-199b-5p, miR-338-3p, miR-486-3p, miR-486-5p, miR-500, miR-502-3p, miR-548	miR-199a-5p, miR-455-3p, miR-577, miR-656
Mor et al. (2015) [76]	24 (12/12)	Brain tissue samples taken from postmortem Brodmann’s area 10	miR-7-5p, miR-19a-3p, miR-19b-3p, miR-21-3p, miR-21-5p, miR-142-3p, miR-142-5p, miR-144-3p, miR-146a-5p, miR-155-5p, miR-219-5p, miR-338-5p, miR-379-5p, miR-451a, miR-494, miR-3168	miR-34a-5p, miR-92b-3p, miR-211-5p, miR-3960
Ander et al. (2015) [77]	18 (10/8)	Brain tissue samples taken from postmortem Brodmann’s areas 22, 41, 42	miR-664-3p, miR-4709-3p, miR-4753-5p	miR-1, miR-297, miR-4742-3p
Wu et al. (2016) [78]	56 (28/28)	Tissue samples taken from postmortem cerebellar cortex, Brodmann area 9	miR-10a-5p, miR-18b-5p, miR-20b-5p, miR-21-3p, miR-23a-3p, miR-107, miR-129-2-3p, miR-130b-5p, miR-148a-3p, miR-155-5p, miR-218-2-3p, miR-221-3p, miR-223-3p, miR-335-3p, miR-363-3p, miR-424-3p, miR-424-5p, miR-425-3p, miR-449b-5p, miR-450b-5p, miR-484, miR-629-5p, miR-651-5p, miR-708-5p, miR-766-3p, miR-874-3p, miR-887-3p, miR-940, miR-1277-3p, miR-3938, miR-2277-5p, let-7g-3p	miR-204-3p, miR-491-5p, miR-619-5p, miR-3687, miR-5096
Huang et al. (2015) [79]	40 (20/20)	Peripheral blood sample taken, microarray (5 ASD/5 controls), and quantitative Real-Time PCR (15 ASD/15 controls)	miR-34b-3p, miR-34c-3p, miR-483-5p, miR-494, miR-564, miR-642a-3p, miR-574-5p, miR-575, miR-921, miR-1246, miR-1249, miR-1273c, miR-4270, miR-4299, miR-4436a, miR-4443, miR-4516, miR-4669, miR-4721, miR-4728-5p, miR-4788, miR-5739, miR-6086, miR-6125	miR-15a-5p, miR-15b-5p, miR-16-5p, miR-19b-3p, miR-20a-5p, miR-92a-3p, miR-103a-3p, miR-195-5p, miR-451a, miR-574-3p, miR-940, miR-1228-3p, miR-3613-3p, miR-3935, miR-4436b-5p, miR-4665-5p, miR-4700-3p, let-7a-5p, let-7d-5p, let-7f-5p
Toma et al. (2015) [80]	1309 (636/673)	Genomic DNA isolated from blood lymphocytes, or from saliva	miR-133b/miR-206 cluster; pooled analysis: miR-133b/miR-206 and miR-17/miR-18a/miR-19a/miR-20a/miR- 19b-1/miR92a-1.	N/A
Hicks et al. (2016) [81]	45 (24/21)	Salivary samples	miR-7-5p, miR-28-5p, miR-127-3p, miR-140-3p, miR-191-5p, miR-218-5p, miR-335-3p, miR-628-5p, miR-2467-5p, miR-3529-3p	miR-23a-3p, miR-27a-3p, miR-30e-5p, miR-32-5p
Nguyen et al. (2016) [82]	14 (8/6)	Samples taken from olfactory mucosal stem cells and skin fibroblasts or Peripheral Blood Mononuclear Cells. Measured through microarray and quantitative Real-Time PCR validation	miR-146a	miR-221, miR-654-5p, miR-656
Kichukova et al. (2017) [83]	60 (30/30)	Blood samples. Quantitative Real-Time PCR validation	miR-18b-3p, miR-106b-5p, miR-142-3p, miR-210-5p, miR-365a-3p, miR-374b-5p, miR-619-5p, miR-664a-3p, miR-3620-3p, miR-4489, miR-8052	hsa-let-7i-3p, miR -15a-5p, miR -20b-3p, miR -29c-5p, miR -96-5p, miR -145-5p, miR -183-5p, miR -193b-3p, miR -197-5p, miR-199a-5p, miR -301a-3p, miR -328-3p, miR -424-5p, miR -486-3p, miR -487b-3p, miR -500a-5p, miR -504-5p, miR -576-5p, miR -587-3p, miR-589-3p, miR -664b-3p, miR -671-3p, miR -3064-5p, miR -3135a, miR -3674, miR -3687, miR-3909, miR -6799-3p, miR -6849-3p
Jyonouchi et al. (2017) [84]	96 (69/27)	Peripheral blood monocytes samples, miRNA expression determined by high-throughput sequencing	hsa-let-7a-1, hsa-let-7a-2, hsa-let-7a-3, hsa-let-7f-1, hsa-let-7f-2, hsa-let-7g, hsa-let-7i, miR-17, miR-26a-2, miR-30b, miR-30c-1, miR-30c-2, miR-98, miR-106b, miR-130a, miR-148a, miR-148b, miR-150, miR-186, miR-301a, miR-374b, miR-494, miR-1248, miR-3607, miR-3609	hsa-let-7b, miR-15a, miR-15b, miR-16-1, miR-16-2, miR-18a, miR-19a, miR-19b-1, miR-19b-2, miR-20a, miR-21, miR-27a, miR-27b, miR-29a, miR-29b-1, miR-29b-2, miR-29c, miR-30e, miR-93, miR-101-1, miR-101-2, miR-103a-1, miR-103a-2, miR-107, miR-126, miR-142, miR-145, miR-146a, miR-151a, miR-181a-1, miR-181a-2, miR-199b, miR-221, miR-222, miR-320a, miR-376c, miR-409, miR-423, miR-484, miR-625, miR-4433b, miR-5701-1, miR-5701-2
Pagan et al. (2017) [85]	517 (239/278) *	Post-mortem pineal glands (melatonin) in 9 patients and 22 controls; gut samples (serotonin) in 11 patients and 13 controls; blood platelets from 239 individuals with ASD, their first-degree relatives and 278 controls	Plasmatic and pineal miR-451	N/A
Nguyen et al. (2018) [86]	11 (5/6)	Post-mortem analysis of temporal lobe in ASD children and controls, miRNA expression performed using Taqman assay	miR-146a	N/A
Yu et al. (2018) [87]	43 (20/23)	Serum samples, quantitative reverse transcription-PCR to examine miRNAs	miR-486-3p, miR-557	N/A
Williams et al. (2018) [88]	128 (48/80) *	Blood samples from 48 ASD and 80 parents	miR-873-5p	N/A

**Table 2 jcm-08-00088-t002:** MicroRNAs in atopic dermatitis (*control cohort represented by patients with early-stage mycosis fungoides (MF1)).

Study	*N* (Case/Control)	Design	Findings	
			Up-Regulated miRNA	Down-Regulated miRNA
Sonkoly et al. (2010) [66]	47 (18/29)	Skin samples	miR-155	
Lv et al. (2014) [89]	58 (30/28)	Serum and urine samples	miR-203, miR-483-5p (serum)	miR-203 (urine)
Ralfkiaer et al. (2014) [90]	75 (20/55) *	Skin samples	miR-149, miR-Plus-C1070, miR-205, miR-141, miR-23b, miR-221, miR-27b, miR-203, miR-7b, miR-19b, miR-27a, miR-455-3p, miR-200a, miR-211, miR-23a, miR-214	miR-181a, miR-342-5p, miR-766, miR-7i, miR-186, miR-342-3p, miR-664, miR-425, miR-9, miR-331-3p, miR-146b-5p, miR-10a, miR-663, miR-937, miR-361-3p, miR-605, miR-146a, miR-940, miR-150, miR-1913, miR-155, miR-302c
Rebane et al. (2014) [91]	18 (9/9)	Skin samples	miR-146a	
Ma et al. (2015) [92]	64 (33/31)	Skin samples	miR-155	
Ding et al. (2016) [93]	22 (14/8)	Skin samples	miR-148b, miR-152, miR-324	
Yang et al. (2017) [94]	37 (37/0)	Skin samples		miR-124

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
