# Peer review of "MicroRNA Cross-Involvement in Autism Spectrum Disorders and Atopic Dermatitis: A Literature Review"

_jcm, 2019, doi:10.3390/jcm8010088_

Round 1
Reviewer 1 Report
There is no research evidence that the incidence of autism is increasing (abstract). The prevalence is.
Good Introduction giving the background for miRNA as a focus for this study.
Used standard of systematic literature review, under the PRISMA guidelines and strategies are described well.
Results are clearly laid out in Tables 1 and 2.
The Discussion of the gene expression association is interesting and does not go beyond the data.
The association of several miRNA in autism and atopic dermatitis is interesting but does not have clinical significance at this point. There is appropriate speculation that they could serve as a biomarker for either disorder or even a treatment target in the future.
Author Response
Thank you for your revision. Please, find below our point-by-point response (in Italics) to your comments.
There is no research evidence that the incidence of autism is increasing (abstract). The prevalence is.
Thank you for your comment. You are completely right. We changed the sentence in the Abstract accordingly.
Good Introduction giving the background for miRNA as a focus for this study.
Used standard of systematic literature review, under the PRISMA guidelines and strategies are described well.
Results are clearly laid out in Tables 1 and 2.
The Discussion of the gene expression association is interesting and does not go beyond the data.
The association of several miRNA in autism and atopic dermatitis is interesting but does not have clinical significance at this point. There is appropriate speculation that they could serve as a biomarker for either disorder or even a treatment target in the future.
Thank You for Your comments and for Your revision.
Reviewer 2 Report
Tonacci et al. aim to find a common genetic link between Autism spectrum disorder (ASD) and Atopic dermatitis (AD) considering miRNA genes previously implicated in each disorder; the authors report miR-146 and miR-155 for having potential pleiotropic effects in both disorders. Recently, there is an increased attention about the neuroinflammatory hypothesis in ASD, and this study may add additional information and links for future inflammatory approach in ASD. This review contributes to the field, but has several important gaps in the genetics of ASD. The review cannot be recommended for publication as it is now, although is generally well written. My main concerns or suggestions to address are:
1) The authors are also authors or two other reviews about ASD and AD, but there is not much work, a part their reviews, about the evidence that is in common between the two disorders. The authors do not explain why they ended up focusing in two apparently not correlated diseases. The link between the two disorders is explained by the increased prevalence in the two disorders, which is extremely concerning. There is no clear evidence that the prevalence in ASD is increased, and the increased numbers are explained by experts in epidemiology of ASD with an improved detection and expanded definition of ASD. Thus, I would be cautious stating that ASD is increasing, and use this hypothesis to correlate ASD and AD, which looks very weak if explained in this way. The authors should find more objective reasons, for instance follow the inflammation hypothesis.
In regards to the prevalence of ASD the authors state: “Their prevalence has dramatically increased in last decades, from 4/10,000 in 2008 [3] to 1/68 cases”, which suggest that nowadays ASD is even worse than a viral pandemic infection. Please correct these numbers referring to the studies of Eric Fombonne, who is the major expert in the epidemiology of ASD (i.e. Eric Fombonne, Editorial: The rising prevalence of autism, 2018; Chakrabarti, S. & Fombonne, E., 2005, American Journal of Psychiatry).
2) The publications listed in Table 1 about miRNAs implicated in ASD include only gene expression studies, although the authors refer to them as genetic studies; relevant genetic studies are not included. The review should include the main additional studies about the evidence of genetic variants within miRNA genes associated or suggested to be implicated in ASD. The most relevant are: the first genetic study (sequencing and association study) of miRNAs in ASD (Common and rare variants of microRNA genes in autism spectrum disorders, 2015; World J Biol Psychiatry), several additional studies published later such as: a) An integrative analysis of non-coding regulatory DNA variations associated with autism spectrum disorder; 2018, Mol Psychiatry; b) Disruption of melatonin synthesis is associated with impaired 14-3-3 and miR-451 levels in patients with autism spectrum disorders; 2017, Sci Rep. Re-consider the review after these additions.
Following here some minor comment:
3) The revision in the Diagnostic and Statistical Manual of Mental Disorders version 5 (DSM-5), has made Autism spectrum disorder as unique diagnostic category, thus should be spelled in singular and not in plural.
4) Studies suggest that heritability in ASD is estimated to be around 80% (and not 50%), here the most updated and relevant references: Colvert et al.(2015). Heritability of autism spectrum disorder in a UK population-based twin sample. JAMA Psychiatry 72, 415–423; Sullivan, et al. (2012). Genetic architectures of psychiatric disorders: The emerging picture and its implications. Nature Reviews Genetics, 13, 537–551; Sullivan, et al (2012). Genetic architectures of psychiatric disorders: The emerging picture and its implications. Nature Reviews Genetics, 13, 537–551.
5) Please when refer to a study cite the relevant research work/s and not the review (ref. 9); that is the case for the de novo variants. Along with the de novo variants should be expanded the part about the inherited rare variants that authors commented briefly, citing the main work (for instance about the exome or sequencing studies in multiplex autism families).
6) The Table legends should be embedded before the tables
Author Response
Thank you for your revision. Please, find below our point-by-point response (in Italics) to your comments.
Tonacci et al. aim to find a common genetic link between Autism spectrum disorder (ASD) and Atopic dermatitis (AD) considering miRNA genes previously implicated in each disorder; the authors report miR-146 and miR-155 for having potential pleiotropic effects in both disorders. Recently, there is an increased attention about the neuroinflammatory hypothesis in ASD, and this study may add additional information and links for future inflammatory approach in ASD. This review contributes to the field, but has several important gaps in the genetics of ASD. The review cannot be recommended for publication as it is now, although is generally well written. My main concerns or suggestions to address are:
1) The authors are also authors or two other reviews about ASD and AD, but there is not much work, a part their reviews, about the evidence that is in common between the two disorders. The authors do not explain why they ended up focusing in two apparently not correlated diseases. The link between the two disorders is explained by the increased prevalence in the two disorders, which is extremely concerning. There is no clear evidence that the prevalence in ASD is increased, and the increased numbers are explained by experts in epidemiology of ASD with an improved detection and expanded definition of ASD. Thus, I would be cautious stating that ASD is increasing, and use this hypothesis to correlate ASD and AD, which looks very weak if explained in this way. The authors should find more objective reasons, for instance follow the inflammation hypothesis.
Thank You for Your useful and interesting comment. We modified the Introduction adding some evidences about the immune involvement in ASD, thus trying to clarify the rationale for our hypothesized association.
In regards to the prevalence of ASD the authors state: “Their prevalence has dramatically increased in last decades, from 4/10,000 in 2008 [3] to 1/68 cases”, which suggest that nowadays ASD is even worse than a viral pandemic infection. Please correct these numbers referring to the studies of Eric Fombonne, who is the major expert in the epidemiology of ASD (i.e. Eric Fombonne, Editorial: The rising prevalence of autism, 2018; Chakrabarti, S. & Fombonne, E., 2005, American Journal of Psychiatry).
Thank You. We modified the sentence in the Introduction section and added the references suggested.
2) The publications listed in Table 1 about miRNAs implicated in ASD include only gene expression studies, although the authors refer to them as genetic studies; relevant genetic studies are not included. The review should include the main additional studies about the evidence of genetic variants within miRNA genes associated or suggested to be implicated in ASD. The most relevant are: the first genetic study (sequencing and association study) of miRNAs in ASD (Common and rare variants of microRNA genes in autism spectrum disorders, 2015; World J Biol Psychiatry), several additional studies published later such as: a) An integrative analysis of non-coding regulatory DNA variations associated with autism spectrum disorder; 2018, Mol Psychiatry; b) Disruption of melatonin synthesis is associated with impaired 14-3-3 and miR-451 levels in patients with autism spectrum disorders; 2017, Sci Rep. Re-consider the review after these additions.
We totally agree with you. We added the publications you cited in Table 1 as this part was (mistakenly) not faced in the previous version of our review. However, taking into account the results of such works, and considered the evidences retrieved by the works already included in the previous version of the review, the miRNAs more widely concerned in both the conditions remain miR-146a and miR-155, as mentioned.
Following here some minor comment:
3) The revision in the Diagnostic and Statistical Manual of Mental Disorders version 5 (DSM-5), has made Autism spectrum disorder as unique diagnostic category, thus should be spelled in singular and not in plural.
Thank You. We corrected accordingly throughout the text.
4) Studies suggest that heritability in ASD is estimated to be around 80% (and not 50%), here the most updated and relevant references: Colvert et al.(2015). Heritability of autism spectrum disorder in a UK population-based twin sample. JAMA Psychiatry 72, 415–423; Sullivan, et al. (2012). Genetic architectures of psychiatric disorders: The emerging picture and its implications. Nature Reviews Genetics, 13, 537–551; Sullivan, et al (2012). Genetic architectures of psychiatric disorders: The emerging picture and its implications. Nature Reviews Genetics, 13, 537–551.
Thank You. We modified the percentage and added the relevant references You suggested us.
5) Please when refer to a study cite the relevant research work/s and not the review (ref. 9); that is the case for the de novo variants. Along with the de novo variants should be expanded the part about the inherited rare variants that authors commented briefly, citing the main work (for instance about the exome or sequencing studies in multiplex autism families).
Thank You. In addition to the review by Vorstman et al., we added the reference to a relevant original article dealing with the analysis of de novo variants. Concerning both de novo and inherited rare variants, we added the interesting article by Al-Mubarak and colleagues (Sci.Rep. 2007) that, conducted on an Arab population, provides significant insights in this specific topic faced within ASD, with potentially useful cues related to the immune engagement in ASD. Such cues were briefly mentioned in the new version of our review.
6) The Table legends should be embedded before the tables
Thank You. We moved the legends before the tables.
Round 2
Reviewer 2 Report
The authors amended successfully most of the criticisms; unfortunately, I must highlight additional concerning to this reviewed version:
1) The authors still explain the association between ASD and AD with the increasing prevalence (or incidence) which is stated in the abstract: “incidence (prevalence of ASD) worldwide is dramatically increasing”. That statement is not correct. In my previous comments I highlighted with several references that there are not reliable data suggesting that ASD prevalence is increasing, nor the incidence (which increases if prevalence increase; they are two measures of the same event). As reported by Baxter et al. (The epidemiology and global burden of autism spectrum disorders, Psychol Med, 2015): “After accounting for methodological variations, there was no clear evidence of a change in prevalence for autistic disorder or other ASDs between 1990 and 2010. Worldwide, there was little regional variation in the prevalence of ASDs.” The authors should remove all statements that are not supported by clear evidence regarding the increase of ASD cases regarding the prevalence or incidence (in abstract and along the paper).
2) Regarding the references about the inherited rare variants, the authors should cite the main manuscripts performed in autism sequencing; the works that represent milestones in the research of this field. The work performed by Al-mubarak et al. (Ref 15), may be useful and can be cited, but is performed in trios, and not in multiplex families (which are the kind of genetic studies that are used to underlie the pattern of inheritance for rare variants). The main works of sequencing in multiplex autism families are these: the first WES in multiplex autism families (Exome sequencing in multiplex autism families suggests a major role for heterozygous truncating mutations. PMID: 25903372; Mol Psychiatry; 2014) and the first WGS in multiplex autism families (Whole-genome sequencing of quartet families with autism spectrum disorder. PMID: 25621899; Nat Med; 2015). Also consider that the rate for coding De novo (DN) variants in ASD is < 1 per proband, and not all of them are pathogenic, which means that the contribution of DN are very limited compared to inherited rare variants. Thus, the sentence “the interplay between de novo and inherited rare variants could be pivotal in the development of ASD” should be rearranged considering the reduced impact of DN in ASD, which explain around 2-5% of the genetic variance in the liability to ASD.
Author Response
Thank you for your revision. Please, find below our point-by-point response (in Italics) to your comments.
1) The authors still explain the association between ASD and AD with the increasing prevalence (or incidence) which is stated in the abstract: “incidence (prevalence of ASD) worldwide is dramatically increasing”. That statement is not correct. In my previous comments I highlighted with several references that there are not reliable data suggesting that ASD prevalence is increasing, nor the incidence (which increases if prevalence increase; they are two measures of the same event). As reported by Baxter et al. (The epidemiology and global burden of autism spectrum disorders, Psychol Med, 2015): “After accounting for methodological variations, there was no clear evidence of a change in prevalence for autistic disorder or other ASDs between 1990 and 2010. Worldwide, there was little regional variation in the prevalence of ASDs.” The authors should remove all statements that are not supported by clear evidence regarding the increase of ASD cases regarding the prevalence or incidence (in abstract and along the paper).
Thank You for Your suggestion. We removed all the statements pertaining prevalence and incidence data both in the Abstract and throughout the text.
2) Regarding the references about the inherited rare variants, the authors should cite the main manuscripts performed in autism sequencing; the works that represent milestones in the research of this field. The work performed by Al-mubarak et al. (Ref 15), may be useful and can be cited, but is performed in trios, and not in multiplex families (which are the kind of genetic studies that are used to underlie the pattern of inheritance for rare variants). The main works of sequencing in multiplex autism families are these: the first WES in multiplex autism families (Exome sequencing in multiplex autism families suggests a major role for heterozygous truncating mutations. PMID: 25903372; Mol Psychiatry; 2014) and the first WGS in multiplex autism families (Whole-genome sequencing of quartet families with autism spectrum disorder. PMID: 25621899; Nat Med; 2015). Also consider that the rate for coding De novo (DN) variants in ASD is < 1 per proband, and not all of them are pathogenic, which means that the contribution of DN are very limited compared to inherited rare variants. Thus, the sentence “the interplay between de novo and inherited rare variants could be pivotal in the development of ASD” should be rearranged considering the reduced impact of DN in ASD, which explain around 2-5% of the genetic variance in the liability to ASD.
We completely agree with You, and thank You for Your useful comment. We modified the sentence about inherited rare variants and de novo variants as suggested and added the most relevant references You cited.